# Evolutionary and Structural Analysis of the Aquaporin Gene Family in Rice

**DOI:** 10.3390/plants14132035

**Published:** 2025-07-03

**Authors:** Tao Tong, Fanrong Zeng, Shuzhen Ye, Zhijuan Ji, Yanli Wang, Zhong-Hua Chen, Younan Ouyang

**Affiliations:** 1China National Rice Research Institute, Hangzhou 311401, China; tongtao@caas.cn (T.T.); yeshuzhen@caas.cn (S.Y.); jizhijuan@caas.cn (Z.J.); 2MARA Key Laboratory of Sustainable Crop Production in the Middle Reaches of the Yangtze River (Co-Construction by Ministry and Province), College of Agriculture, Yangtze University, Jingzhou 434025, China; fanrong.zeng@yangtzeu.edu.cn; 3State Key Laboratory for Quality and Safety of Agro-Products, Key Laboratory of Agricultural Microbiome of Zhejiang Province, Key Laboratory of Biotechnology in Plant Protection of MARA, Institute of Plant Protection and Microbiology, Zhejiang Academy of Agricultural Sciences, Hangzhou 310021, China; ylwang88@aliyun.com; 4School of Science, Western Sydney University, Penrith, NSW 2751, Australia; z.chen@westernsydney.edu.au; 5Hawkesbury Institute for the Environment, Western Sydney University, Penrith, NSW 2751, Australia

**Keywords:** aquaporin, rice, functional plasticity and evolution, stress adaptation mechanisms, transmembrane channel proteins, water and solute transport

## Abstract

Aquaporins in rice (*Oryza sativa* L.) represent a pivotal class of transmembrane channel proteins that mediate the bidirectional transport of water and small solutes, which have critical functions in cellular osmoregulation and ion homeostasis maintenance. Their evolutionary diversity and functional plasticity constitute fundamental mechanisms underlying the adaptive responses to diversified environmental challenges. This review systematically summarizes rice AQPs’ evolutionary origins, structural characteristics, and spatiotemporal expression patterns under both physiological and stress conditions, highlighting the high conservation of their key functional domains across evolution and their environment-driven functional diversification. The molecular mechanisms governing AQPs in water utilization, nutrient uptake, and stress responses are unraveled. Furthermore, the potential of precision gene editing and multi-omics integration is discussed to decipher the intricate relationships between AQP evolutionary history, environmental adaptability, and functional specialization, thereby providing a theoretical basis for advancing crop stress resistance and high-quality breeding.

## 1. Introduction

Aquaporins (AQPs) are integral membrane proteins belonging to the major intrinsic protein (MIP) superfamily, facilitating the transport of water and small neutral solutes across cell membranes in plants [1,2,3]. Based on their sequence homology and subcellular localization, plant AQPs are categorized into several subfamilies: plasma membrane intrinsic proteins (PIPs), tonoplast intrinsic proteins (TIPs), NOD26-like intrinsic proteins (NIPs), small basic intrinsic proteins (SIPs), and the recently identified X intrinsic proteins (XIPs) [4,5].

In plants, AQPs are involved in various physiological processes, including maintaining cellular water balance, regulating ionic homeostasis, and transporting small molecules like glycerol and ammonia [2,6,7,8,9]. In plants experiencing adverse environmental conditions, changes in AQP activity and expression can significantly affect root water transport properties, nutrient acquisition, and transpiration [10]. Simultaneously, the transport activity of AQPs is regulated by various factors, such as phosphorylation, cytosolic pH, divalent cations, and reactive oxygen species, which further fine-tune their functions in plant physiology [4].

Beyond canonical transport roles, AQPs serve as critical PPI (protein–protein interactions) hubs regulating cellular communication. Construction of the PPI interactome and mining of interaction databases demonstrate that AQPs interact with a wide range of proteins, including nutrient transporters, stress-responsive elements, vesicle membrane fusion proteins, and kinases, highlighting their central role in plant physiological processes [11,12,13]. This interactome demonstrates that OsPIPs function as integrative platforms for diverse transport activities and reveals novel regulatory mechanisms of OsPIP cellular trafficking under osmotic and oxidative conditions [11]. Additionally, structural predictions suggest the feasibility of identifying AQP interactions based on structural information alone, revealing that single-nucleotide polymorphisms (SNPs) associated with specific traits could influence the plant interactome [12]. These findings collectively indicate that AQPs not only play pivotal roles in water homeostasis but also serve as key modulators in broader cellular signaling and response pathways. This expanded understanding of aquaporin function provides new perspectives on their functional diversity and regulatory complexity in plants.

Rice is a staple food crop globally, the productivity of which is largely determined by the water supply [14]. Early studies have shown that AQPs are major determinants of water-use efficiency (WUE) in rice, particularly under field conditions [15]. The expression of *AQPs* in the root endodermis is crucial for water uptake, and inadequate expression profiles can limit WUE [15,16]. Additionally, specific expression profiles of *AQPs* have been demonstrated to play a distinct role in the grain-filling process of rice [17]. AQPs exhibiting diversified isoforms reflect their adaptation to various environmental conditions and physiological needs. Research has demonstrated that the enhanced tolerance to environmental stresses of rice plants is related to increased AQP expression and activity, which improves osmotic water fluxes and osmotic adjustment [1]. For example, under both drought and chilling stress, aquaporin gene expression is significantly upregulated. Particularly, the expression of *OsPIP2;5* increased more than fourfold. Further research revealed that this elevated aquaporin protein expression shows a significant positive correlation with drought tolerance and cold resistance [18,19]. This finding suggested that manipulating *AQP* expression could be a viable strategy for developing resistant rice varieties [20].

Plant AQPs have evolved specialized protein architectures through adaptive evolutionary processes, with structural divergence in their selective filters driving functional diversification of substrate transport profiles, thereby conferring molecular plasticity for plants to mitigate environmental stresses [8,20,21,22]. Rice, serving dual roles as a monocotyledonous model organism and a globally critical food crop, exhibits unique structure–function correlations within its AQP family, shaped by dual selective pressures from natural speciation and artificial domestication [23]. These evolutionary adaptations establish a distinctive paradigm for elucidating plant environmental resilience mechanisms. This review systematically synthesizes the multifaceted biological functions of AQPs in rice developmental regulation and stress-responsive pathways. Through integrated analysis of their evolutionary origins, protein structural features, and spatiotemporal expression dynamics, we propose structurally governed functional traits of AQPs shaped by genetic selection and environmental stress. These approaches aim to overcome current bottlenecks in synchronously improving crop water-use efficiency and multi-stress resilience, thereby providing a theoretical framework for next-generation crop engineering under escalating climatic challenges.

## 2. Evolutionary Dynamics of Aquaporin Genes in Rice

### 2.1. The Origin of Aquaporin in Green Plants

Phylogenetic analysis revealed that plant AQPs exhibited a complex evolutionary history, enabling the tracing of their origins in green plants [20]. Putative orthologous protein sequences of AtANN1 were retrieved from 1000 plant transcriptome databases (OneKP) through the China National GeneBank (CNGB) platform (https://db.cngb.org/blast/blast/blastp/?project=onekp; accessed on 1 May 2025), using representative rice aquaporin sequences (OsNIP1;1, OsSIP1;1, OsPIP1;1, and OsTIP1;1) as a reference (Figure 1; Appendix A) [24,25,26]. Orthologous proteins of these four rice AQPs were identified in approximately 75% of the 1322 terrestrial plant and algal species within the OneKP database, enabling comprehensive phylogenetic tree reconstruction.

Evolutionary analyses demonstrated that these four rice AQPs originated from green algae, but their distinct functional localizations and orientations have shaped differential evolutionary trajectories (Figure 1). Notably, OsNIP1;1 exhibited the most divergent evolutionary process compared to the other three proteins. Phylogenetic analysis revealed that its algal homologs share higher sequence similarity with flowering plant NIP1s than with homologs from bryophytes and pteridophytes. This pattern suggests a complex evolutionary history for OsNIP1;1, potentially involving lineage-specific gene retention, loss, or divergence events after the separation of major plant lineages (Figure 1A). Previous studies proposed that terrestrial plant NIPs were likely acquired through horizontal gene transfer from ancestral bacteria or algae [27]. Given their primary role in regulating water and metabolite flux between roots and nitrogen-fixing bacteria, the unique evolutionary trajectory of NIP proteins might reflect their specialized functional outcomes [28,29].

PIPs, involved in CO_2_ and water transport across organisms from bacteria to higher plants, displayed relatively short evolutionary branches in terrestrial plants (Figure 1B), suggesting their conserved nature and low genetic divergence—features likely associated with their critical physiological roles [30,31]. Current phylogenetic analyses across multiple plant species consistently identify TIPs and PIPs as separate, well-supported subfamilies. While some analyses suggest TIPs could have derived from PIP-like ancestors, the phylogenetic data do not unequivocally confirm this relationship, and TIPs maintain distinct sequence motifs and selectivity filters compared to PIPs (Figure 1C) [4,22,32]. SIPs (small endoplasmic reticulum-localized intrinsic proteins) exhibited structurally and functionally differences from other MIPs. They possess unique ar/R selectivity filters and motif patterns, setting them apart from TIPs, PIPs, and other subfamilies [33,34,35,36]. Notably, SIPs are found across a wide range of plant lineages, from basal to advanced species, with a clear, separate evolutionary trajectory, supporting their status as an ancient and conserved group (Figure 1D) [37,38].

Collectively, rice AQPs demonstrated high evolutionary conservation while optimizing their evolutionary trajectories through functional characteristics to adapt to diverse environmental conditions.

### 2.2. Gene Expansion and Diversification of Aquaporin in Rice and Other Plants

To further investigate the gene expansion and diversification processes of the plant AQP family, we employed a Hidden Markov Model (HMM) strategy to search for AQP sequences from the local hmmsearch retrieval (http://hmmer.org/; 26 November 2020 releases) and SUPERFAMILY database (https://supfam.mrc-lmb.cam.ac.uk/SUPERFAMILY/index.html; accessed on 1 May 2025) [39]. A total of 1119 candidate AQP family members were identified across 24 representative algal and terrestrial plant lineages. Phylogenetic analysis revealed that plant AQPs could be categorized into several subclasses, each containing multiple evolutionary clades. Notably, the evolutionary pattern of AQPs did not strictly follow species phylogeny but rather appears to be driven by functional demands, suggesting that ancient and conserved AQPs may persist in dicotyledonous plants (Figure 2).

The phylogenetic branches demonstrated the existence of dozens of orthologous protein pairs between species, such as *Prunus persica* (ppa009691m) and *Fragaria vesca* (gene23712), *Carica papaya* (evm.TU.supercontig_20.95) and *Populus trichocarpa* (POPTR_0004s18240.1), and *Physcomitrella patens* (Phypa1_1.68172) and *Solanum lycopersicum* (Solyc05g055990.2.1) (Figure 2). These findings indicated that the expansion of plant *AQPs* has likely not only involved intra-species gene duplication events, though some comparative genomic studies have proposed horizontal gene transfer (HGT) between species as an additional diversification mechanism [27]. While HGT remains a debated hypothesis in plant evolution and environmental adaptability, its potential role in AQP diversification requires validation through phylogenomic synteny analyses [40,41,42,43,44]. Specifically, in rice, gene duplication events have played a crucial role in shaping the AQP gene family. Tandem duplications serve as a significant mechanism for expanding this gene family. Moreover, whole-genome duplication (WGD) events have contributed to the diversification and functional specialization of *AQPs*, enhancing rice’s adaptability to various environmental stresses [20,45]. Sequence alignments and homology analyses further indicate that point mutations and small-scale duplications have fine-tuned the functions of individual aquaporins, allowing for specialized roles in water transport, nutrient uptake, and stress responses [20]. Additionally, chromosomal translocation events may have facilitated the relocation of *AQP* genes to different chromosomes, potentially leading to novel regulatory patterns or expression specificities [46]. These genetic events collectively highlight the dynamic evolutionary history of rice aquaporins, contributing to their functional diversity and adaptation to diverse environments.

Previous phylogenetic analyses of AQPs in *Arabidopsis thaliana*, *Glycine max*, *Zea mays*, and *Cicer arietinum* have demonstrated their high species specificity [38,47]. Therefore, the expansion and diversification of *AQP* genes might be associated with plant adaptation to distinct ecological niches and environmental stresses [48,49,50]. This evolutionary pattern further implied that the genetic inheritance and expansion of *AQPs* contribute to the development of distinct agronomic traits in crops such as rice, which may confer adaptive advantages or introduce potential trade-offs under specific environmental contexts.

## 3. Structural Characteristics of Rice Aquaporins

### 3.1. Molecular Architecture and Evolutionary Adaptations Governing Water Transport and Substrate Selectivity

Aquaporins, members of the major intrinsic protein (MIP) family, are evolutionarily conserved transmembrane proteins characterized by six α-helices (TM1–TM6) and five connecting loops [51] (Figure 3). These structural elements formed a homotetrameric channel with cytoplasmic termini, where each monomer independently facilitates water transport. Central to proton exclusion were two conserved asparagine–proline–alanine (NPA) motifs embedded in the hydrophobic cores of loops B and E [4,52,53]. The NPA motifs adopted opposing orientations, folding into half-helices that converge to create a bipolar aqueous pore. This unique architecture generated an electrostatic barrier via the dipole moments of the helices, forcing water molecules into a unidirectional orientation that disrupted proton hopping while enabling rapid water permeation (~3 × 10^9^ molecules/s) [54,55,56]. Additionally, the NPA motifs sterically restricted the pore diameter (~3 Å) and mediated hydrogen bonding with water molecules, further ensuring strict proton exclusion [57,58]. Adjacent to the NPA region lays the aromatic/arginine (ar/R) filter, typically including aromatic residues (such as phenylalanine, tyrosine, or histidine) and a conserved arginine, creating a unique environment that combines hydrophobicity and a positive charge. This area is the narrowest point along the channel and functions as a selectivity filter, excluding anything bulkier than water. In aquaglyceroporins, the ar/R-region typically exhibited ∼1 Å width, permitting transport of larger solutes [21,52,59] (Figure 3B). These features ensure efficient water flow, strict exclusion of protons and ions, and the maintenance of cellular water balance.

The conserved molecular architecture of AQPs provides the structural foundation for selective water transport. Concurrently, specific amino acid substitutions and conformational changes in NPA motifs and ar/R filter composition can alter substrate selectivity and transport efficiency, supporting adaptation to different environmental and physiological demands. For instance, in aquaglyceroporins, substitution of the conserved asparagine residue within the second NPA motif to an aspartic residue (NPD variant) structurally reconfigures the channel’s selectivity filter, which expands the pore diameter to accept larger solutes such as glycerol [60]. Compared to AtPIP2;1, the substitution of specific residues in loop C of AtPIP2;3 triggers a conformational shift of the ar/R selectivity filter, ultimately resulting in its lack of CO_2_ permeability [61]. Notably, among plant aquaporins, TIPs display the most pronounced sequence variability within their ar/R selectivity filters. Beyond facilitating water transport, TIP isoforms have demonstrated permeability to multiple essential metabolites, including ammonia/ammonium (NH_3_/NH_4_^+^), urea, glycerol, and H_2_O_2_ [62].

### 3.2. Three-Dimensional Structural Modeling Guides Functional Analysis of Rice Aquaporin

The application of 3D structural modeling has significantly advanced our understanding of functions in rice AQPs [63]. The structures of 11 rice AQPs were predicted with high accuracy using the AlphaFold model [64]. The results revealed that the AQPs in rice exhibit relatively conserved transmembrane domain conformations, while structural and sequence variations were observed among different subfamilies (Figure 3A). These variations underpin functional diversity in tissue localization, substrate selectivity, and regulatory modes among rice AQPs [52]. Furthermore, AlphaFold-based structural modeling has been applied to explore interaction networks of rice AQPs. For example, an *in silico* study was conducted to uncover the interaction between Harpin (Hpa1) and the rice aquaporin OsPIP1;3 [65]. This point was confirmed by experimental evidence from co-immunoprecipitation assays. The interaction between OsPIP1;3 and the bacterial protein Hpa1 disabled the CO_2_-transporting function of OsPIP1;3. This interaction shifts OsPIP1;3 from CO_2_ transport to effector translocation, aggravating bacterial virulence while compromising rice photosynthesis. Critically, inhibiting this interaction through external application of isolated Hpa1 to rice plants could revert OsPIP1;3 to its CO_2_ transport function, abrogating bacterial virulence and enhancing photosynthesis [66,67,68,69]. The methodology established in these studies represents a practical and feasible approach for investigating the structure–function relationships of proteins. Generally, establishing the structural model of rice aquaporins provides critical guidance for both functional investigation and practical applications of these proteins.

## 4. Functional Roles of Aquaporins in Rice Physiology

### 4.1. Aquaporin-Mediated Water Homeostasis and Nutrient Transport in Rice

Water uptake and transport constitute fundamental processes underpinning plant growth and development. Aquaporins, a class of selective transmembrane channel proteins localized on cellular membranes, mediated highly efficient water molecule transport in [5]. In 1992, researchers utilizing the *Xenopus laevis* oocyte heterologous expression system demonstrated the water transport activity of the membrane protein CHIP28, subsequently designated as Aquaporin-1 (AQP1). This groundbreaking study first revealed the existence of protein-mediated transmembrane water transport pathways in biological membranes [70,71]. The aquaporin r-TIP was previously isolated from *Arabidopsis thaliana*, and its water transport function was confirmed, marking it out as the first reported plant AQP [72].

Current research indicates that, in plants, PIPs and TIPs constitute the predominant AQP families, primarily regulating water flux at cellular and subcellular levels. PIPs, predominantly localized to the plasma membrane, were classified into PIP1 and PIP2 subfamilies based on sequence divergence at their N-and C-termini [73]. Functional characterization revealed similar physiological roles among PIP isoforms: *OsPIP1;1* and *OsPIP2;1* exhibited upregulation in both roots and leaves to facilitate membrane water diffusion [74]. Notably, overexpression of *OsPIP2;7* in rice enhanced transpiration rate, which indicated that OsPIP2;7 participates in rapid cellular water movement and the maintenance of water homeostasis [75]. Importantly, OsPIP2;4, a root-specific isoform, exhibited bifunctional activity by mediating not only water homeostasis but also selective transport of alkali monovalent cations, suggesting its role in osmoregulatory crosstalk [76]. Furthermore, OsPIP2;5 displayed transpiration-dependent induction and polarized membrane localization, implicating its central role in fine adjustment of radial water transport in roots [77]. Intriguingly, OsPIP1;2 was found to be involved in CO_2_ transport from the leaf intercellular air space to the chloroplast, contributing to CO_2_ assimilation and phloem sucrose transport [65]. Vacuoles harbor a single subclass of AQPs termed TIPs. In rice, several TIP isoforms, OsTIP1;2, OsTIP2;2, OsTIP4;1, and OsTIP5;1, have been identified to possess water transport activity. Additionally, OsTIP1;2, OsTIP3;2, and OsTIP4;1 have also been shown to facilitate glycerol transport, highlighting their multifunctional role in cellular transport processes [78]. NIPs were categorized into three subgroups—NIPI, NIPII, and NIPIII—according to structural variations and substrate specificities. NIPIs primarily transported water, glycerol, and lactic acid, whereas NIPIIs and NIPIIIs specialized in translocating uncharged solutes such as silicic acid, boric acid, arsenite, selenite, and germanic acid [79]. Specifically, OsNIP3;3 was identified as a unique AQP facilitating water, hydrogen peroxide, and arsenite transport [80]. In deepwater rice, the expression of *OsNIP2;2* (silicic acids transporter) and *OsNIP3;1* (boric acids transporter) decreased remarkably for the rapid internode elongation during submersion. Specifically, *OsNIP2;2* downregulation reduces silicic acid deposition in cell walls to facilitate cell expansion, while concurrent *OsNIP3;1* suppression prevents boron toxicity in elongating internodes [81].

### 4.2. Role of Aquaporins in Rice Abiotic Stress Responses

In rice, AQPs have been identified as key players in the response to various abiotic stresses, including drought, salinity, and temperature extremes. Research indicated that PIP family genes in rice exhibited tissue-specific expression patterns and differentiated physiological functions during drought response. OsPIP2;2 has been shown to enhance water transport efficiency, thereby improving drought tolerance in rice. OsPIP2;2 facilitates water uptake and helps maintain cell membrane integrity, making it crucial for sustaining plant growth during water scarcity [82]. Similarly, *OsPIP2;3* is mainly expressed in the roots and has been shown to be substantially upregulated under water-deficit conditions [83]. Moreover, OsPIP2;4 has been demonstrated to possess the capability to regulate water transport and non-selective Na^+^ and K^+^ ion conductance in electrophysiological experiments. Its overexpression in plants enhanced root water absorption efficiency under drought conditions, thereby promoting plant growth [76,84]. The root-predominant *OsPIP2;5* has demonstrated water transport activity in yeast systems [33], while overexpression of *OsPIP2;6* could simultaneously enhance rice resistance to drought, waterlogging, salt stress, and rice blast [85,86]. Importantly, multiple *AQP* genes—including *OsPIP1;3*, *OsPIP2;4*, *OsPIP2;5*, *OsTIP2;1*, and *OsNIP2;1*—have consistently exhibited stronger drought-responsive upregulation in drought-tolerant rice varieties [18]. Furthermore, changes in root hydraulic conductivity, facilitated by AQPs, were integral to the overall hydraulic response of rice to salt and osmotic stress [87]. OsPIP1;1 has been implicated in promoting salt resistance and seed germination, highlighting its role in enhancing rice resilience to salinity stress [88]. Simultaneous editing of three aquaporin-encoding genes (*OsPIP1;1*, *OsPIP1;2*, and *OsPIP1;3*) in rice resulted in reduced photosynthetic efficiency and decreased osmolyte accumulation under saline conditions. The engineered mutants exhibited elevated malondialdehyde levels and disrupted osmotic homeostasis, ultimately leading to impaired plant development during salt stress exposure [89]. The cold acclimation process in rice involves coordinated upregulation of root *AQP* gene expression, particularly *OsPIP2;5*, which was essential for enhancing root water uptake during cold stress in [19]. *OsPIP2;7* expressed in *Xenopus oocytes* enhanced water transport activity, while overexpression of *OsPIP2;7* in rice promoted transpiration and resistance to cold stress [78]. *OsPIP1;3* and *OsPIP2* family members showed distinct expression patterns and functional roles in chilling tolerance by tissue (roots vs. shoots) and by specific isoforms. Overexpression of *OsPIP1;3* in transgenic rice improved chilling tolerance by enhancing water permeability, especially when interacting with OsPIP2 proteins [90,91].

### 4.3. Regulation of Aquaporin Expression in Rice

Understanding the regulation of *AQP* expression in rice can provide insights into improving crop resilience and productivity. We analyzed and generated expression profiles of 36 rice *AQP* genes using RNA-Seq data retrieved from 2786 publicly available rice transcriptome libraries in the Rice RNA-Seq database (http://jixianzhai.org/; accessed on 7 May 2025). The expression profiling revealed distinct transcript abundance patterns of rice AQPs across 13 tissue types and under 11 abiotic stress conditions (Figure 4) [92].

Differential expression of rice AQP genes was most pronounced in root tissues among the examined organs (Figure 4). Notably, *OsPIP1;1, OsPIP1;2, OsPIP1;3, OsPIP2;1*, *OsPIP2;2, OsTIP1;1*, and *OsTIP1;2* exhibited consistently high transcript abundance across multiple organs, including the roots, leaves, seedlings, and stems (Figure 4). This pan-tissue expression pattern suggested their central roles in fundamental water transport and nutrient mobilization. Correspondingly, these genes displayed elevated responsiveness to diverse abiotic stresses, particularly drought, salinity, and osmotic stress, suggesting their potential role in enhancing stress tolerance through modulation of cellular water potential or ion co-transport mechanisms [75,78,89,93]. Notably, OsNIP2;1 (Lsi1), the first characterized silicon transporter in higher plants, exhibited exceptional upregulation under cold stress. Previous studies have demonstrated that its overexpression confers enhanced cold tolerance, aligning with our observed expression dynamics (Figure 4) [94]. In contrast, the SIP subfamily showed minimal responsiveness to environmental stresses, potentially indicating preferential involvement in constitutive physiological processes rather than stress adaptation (Figure 4) [95]. Current research on OsSIPs remains limited, with only a recent study demonstrating that OsSIP1 and OsSIP2 primarily localize to the endoplasmic reticulum, where they facilitate water and H_2_O_2_ transport. *OsSIP1* exhibits broader expression patterns than *OsSIP2* across most tissues and developmental stages. Furthermore, both aquaporins show significant upregulation under various abiotic stresses and in response to diverse hormonal treatments [96].

The distinct expression profiles observed between male and female reproductive tissues further imply the existence of sex-specific regulatory mechanisms governing AQP function. *OsPIP2;6* knockout (*OsPIP2;6*-KO) lines were generated to assess potential compensatory mechanisms among rice aquaporins. RT-qPCR analysis of other *OsPIPs* transcripts demonstrated no significant upregulation in KO plants under either basal conditions or pathogen challenge. Consistent with this, the heightened susceptibility of *OsPIP2;6*-KO rice to *Magnaporthe oryzae* indicates that functional compensation by other aquaporins at the transcriptional level is limited.

AQP expression in rice is dynamically regulated at the protein level in response to drought stress and recovery. Quantitative proteomics in *Oryza sativa* cv. Nipponbare revealed that while many AQPs (e.g., PIP1-1, PIP1-2, PIP2-6, and TIP2-2) decrease in abundance during moderate drought, several isoforms (notably PIP2-1 and PIP2-7) are significantly upregulated under severe water deficit, suggesting a role in adaptation to acute stress. Crucially, the abundance of most drought-responsive AQPs rapidly decreased upon re-watering, highlighting their specific involvement in the stress response phase [97]. Furthermore, the functional outcome of aquaporin modulation is genotype-dependent. Overexpression of *OsPIP2;4* enhanced root hydraulic conductivity and drought tolerance in cultivar Giza178 but not in IR64, underscoring the critical influence of genetic background and inherent physiological traits on aquaporin function during water stress [84].

**Figure 4 plants-14-02035-f004:**
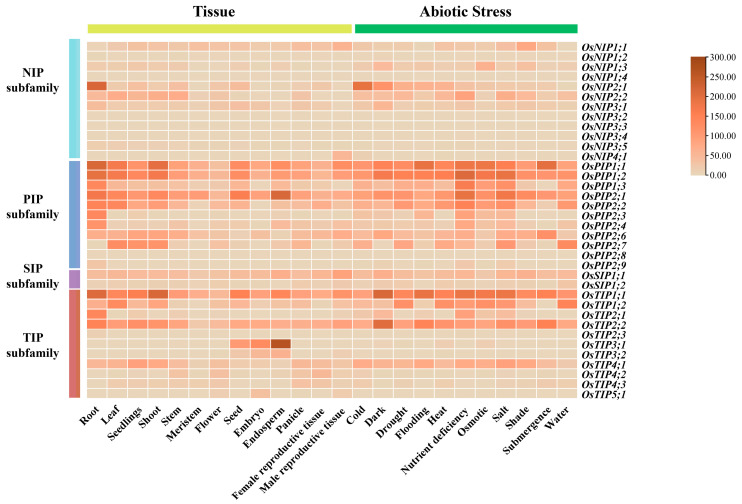
Expression profiles of 36 aquaporin genes in plant tissues and in response to abiotic stresses in rice. The expression data of rice AQP genes were obtained from public plant RNA-seq databases, covering plant tissues and responses to abiotic stress. The heatmap was generated using Tbtools (v2.106) [98]. Gene expression levels were quantified as FPKM values and visualized by color intensity, with dark brown blocks representing high expression and light red blocks representing low expression.

## 5. Future Directions and Challenges in Rice Aquaporin Research

### 5.1. Gene Editing Technology and Functional Dissection Deepening

The breakthrough of gene editing technology provided precision for studying the functions of rice AQPs. The CRISPR/Cas9 system has been successfully applied to rice genome editing, and its highly efficient single-base editing capability made it possible to obtain homozygous mutants within a single generation [99,100]. Researchers revealed the unique roles of specific AQP subfamily members in rice stress resistance through targeted editing. For instance, the single-gene knockout mutants of *OsPIP1;1*, *OsPIP1;2*, and *OsPIP2;1* exhibited significant differences in agronomic traits and photosynthetic–physio-biochemical characteristics. Notably, the *Ospip1;1* mutant demonstrated the lowest tiller number and plant height among all genotypes. With the exception of the *Ospip1;2* mutant, both the *Ospip1;1* and *Ospip2;1* mutants showed markedly reduced tiller numbers and plant heights compared to the wild-type plants [101]. Furthermore, *OsNIP3;1* was also identified as a promising target for modulating arsenic (As) accumulation in rice without compromising grain yield or plant vigor genome editing approaches [102]. There were a large number of *AQP* genes with overlapping functions in rice. The *OsPIP1* gene cluster functioned as critical osmotic regulators essential for enhancing rice salt stress tolerance. Multiplex editing of *OsPIP1;1*, *OsPIP1;2*, and *OsPIP1;3* induced loss-of-function mutations via single T/A base indels, disrupting water transport and osmotic ion homeostasis under salt stress [89]. Functional redundancy in rice *AQPs* might weaken single-gene knockout phenotypes. Future studies should employ multiplex editing strategies to simultaneously target multiple isoforms, effectively overcoming redundancy constraints.

### 5.2. Multi-Omics Integration and Regulatory Network Revelation

The complexity of rice AQP functions required research to shift from the single-gene level to a systems biology perspective. In maize, the integration of genome-wide association studies (GWASs) and expression quantitative trait locus (eQTL) mapping has enabled the identification of key genetic variants and natural diversity regulating *AQPs* expression [50]. Genome-wide expression profiling has helped to identify 36 *AQP* genes in rice, simultaneously revealing their tissue- and developmental stage-specific expression patterns, while comprehensive and precise genetic locus mapping, coupled with expression profiling analyses, remained critically required [4,38]. Proteomic studies further revealed significant changes in the phosphorylation modification levels of multiple AQPs in rice leaves under drought stress, suggesting the important role of post-translational modifications in regulating protein activity [1].

However, the temporal resolution and spatial heterogeneity inherent in multi-omics datasets hindered the elucidation of AQP dynamics [103,104]. Therefore, to comprehensively explore the response characteristics and functional effects of rice AQPs, it is essential to integrate single-cell sequencing technology, spatial transcriptomics, plant electrophysiological techniques, and plant imaging technologies.

### 5.3. Rice Aquaporin Evolution and Structure for Functional Implications

With the rapid development of genome sequencing and transcriptomic technologies, the biological information embedded in plant genomes has become a critical breakthrough for offering opportunities to decipher the intricate mechanisms governing plant responses to environmental stimuli [38,105]. The AQP gene family in rice, comprising four distinct subfamilies—PIP, TIP, NIP, and SIP—likely arose through an adaptive evolutionary strategy driven by the synergistic effects of gene duplication events and functional innovation [20]. This suggests that selective pressures from the external environment drive functional divergence among aquaporin subfamilies; for instance, PIP paralogs were found to be evolved and have specialized roles in CO_2_ transport, while certain NIP subfamily members gained selectivity for arsenite [65,80].

The structural diversity of these family members, shaped during evolution, conferred enhanced substrate selectivity and plasticity in environmental stress responses. Expression profiling further revealed that the coexistence of functional specialization and redundancy enabled rice to adapt to complex or extreme habitats through spatiotemporal-specific regulation. Phylogenetic analyses have revealed a striking correlation between the evolutionary relationships of *AQP* genes and habitat shifts in plants, suggesting that natural selection drives their adaptive evolution by optimizing water-use efficiency [37,106,107,108]. Notably, although rice AQP subfamilies exhibited heterogeneous evolutionary origins (e.g., the PIP subfamily acquired through horizontal gene transfer), their core functional domains (e.g., the NPA motif) remained highly conserved during evolution, indicating the irreplaceable role of these domains in transmembrane water transport [23,27]. Nevertheless, several critical questions remained unresolved, including how the dynamic balance between tissue-specific expression and functional specialization redundancy is established under the interplay of natural selection and environmental stress, the differentiation patterns in stress response characteristics and functional properties among subfamily members, and whether subfamilies form regulatory networks through functional synergy or antagonism [109].

Therefore, future research necessitates the integration of insights from structural biology, evolutionary analysis, and functional genomics. Comprehensive phylogenetic and structural evolution analyses should be leveraged to mine potential functional information of rice AQPs from evolutionary traces. Concurrently, the framework of integrated multi-omics analysis and cooperative multi-gene regulation, precise gene editing techniques, protein functional characterization, and real-time physiological measurements in plants must be considered to elucidate the multifaceted roles of rice AQPs in environmental adaptation, functional execution, and tissue-specific expression. This approach will ultimately provide a theoretical framework and functional basis for crop improvement strategies.

## Figures and Tables

**Figure 1 plants-14-02035-f001:**
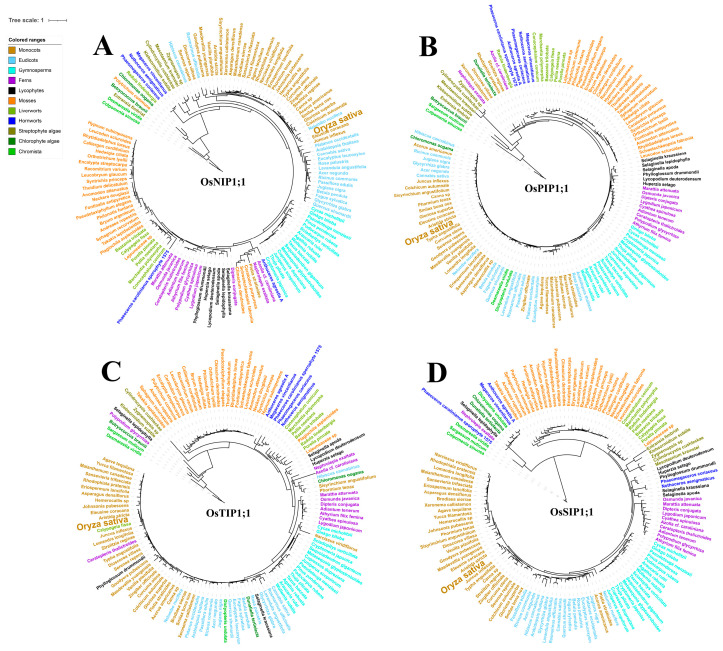
Molecular evolution of representative rice aquaporins in plants and algae. The evolutionary tree of AQPs in rice, representing major green plant lineages, was constructed using the OneKP and NCBI databases. The tree was generated by the maximum likelihood method and the IQ-Tree model, with bootstrap analysis set to 1000 replicates and other parameters at default settings. Overall, 11 evolutionary branches are denoted by distinct colored ranges in the upper-left corner, corresponding to the font colors of their respective clades in the evolutionary tree. The evolutionary relationships of (**A**) OsNIP1;1, (**B**) OsPIP1;1, (**C**) OsTIP1;1, and (**D**) OsSIP1;1 were each constructed using orthologous proteins retrieved from the same set of 135 species.

**Figure 2 plants-14-02035-f002:**
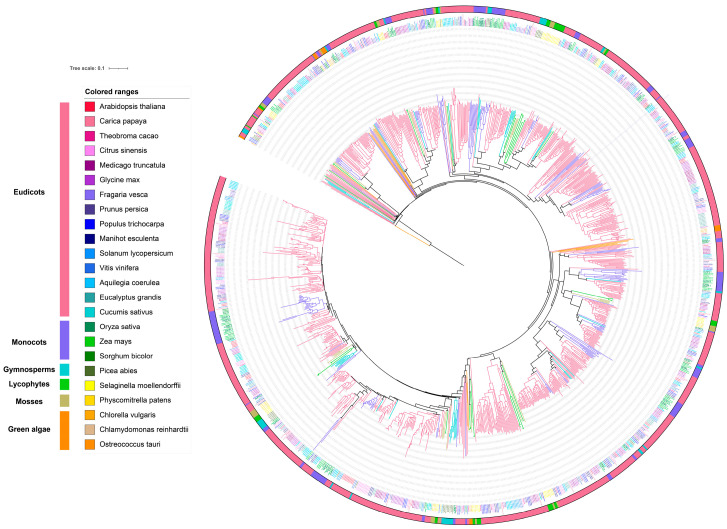
Phylogenetic relationship of canonical aquaporin proteins. The phylogenetic tree was constructed using the maximum likelihood method based on 1119 trimmed AQP amino acid sequences derived from genomic and transcriptomic data obtained from EnsemblPlants and Phytozome databases. The sequences were aligned with MAFFT; the best model was estimated with RAxML. Bootstrap analysis was performed with 1000 replicates while maintaining other parameters at default settings. The multicolored strip on the far left indicates major evolutionary lineages—eudicots, monocots, basal angiosperm, gymnosperm, lycophyte, mosses, and green algae—corresponding to the colored strips on the phylogenetic tree. Overall, 24 representative plant species are listed on the left in phylogenetic order; their colored ranges match the font and clade colors of their respective gene family members in the phylogenetic tree.

**Figure 3 plants-14-02035-f003:**
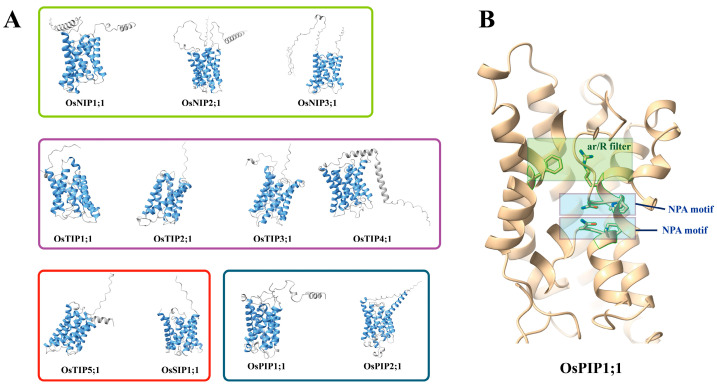
AlphaFold-driven computational modeling of aquaporin structure. (**A**) Three-dimensional structural models of 11 representative rice AQPs were computationally predicted using the AlphaFold intelligent model (https://www.alphafold.ebi.ac.uk/; accessed on 1 May 2025), with transmembrane domains annotated in blue. Distinct colored outlines represent specific aquaporin subfamilies in rice. (**B**) Structural magnification of OsPIP1;1 highlights the NPA motif and ar/R region, demarcated within blue and yellow boxes, respectively. Critical residues are color-coded by blue and red.

## Data Availability

Not applicable.

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
