# Peer review of "Evolutionary and Structural Analysis of the Aquaporin Gene Family in Rice"

_plants, 2025, doi:10.3390/plants14132035_

Round 1

Reviewer 1 Report

Comments and Suggestions for Authors The article delves into the current understanding of root aquaporins, their evolutionary development, and also serves as a review of the available literature, updating certain aspects such as their interactions with other proteins, 3D modeling, involvement in abiotic stresses, and the functional roles of these aquaporins—mostly previously known, but now reinforced by more recent studies. Overall, the article is useful and summarizes important information. However, I would like to highlight several aspects that could improve the manuscript. One recurrent issue observed across all figures is the absence of proper captions. Please urgently include concise legends that indicate what each figure represents, how it was generated, the meaning of each color or symbol, any statistical analyses performed (see Figure 4, for instance), and all relevant information needed for the figure to be understood independently of the main text—just as would be expected in a research article. In the introduction, I recommend adding some information on the ability of aquaporins to interact with other proteins, which is a relatively novel area of study. Some authors propose that aquaporins play a key role in cellular communication or serve as hubs for protein–protein interactions. I suggest reviewing and including insights from the following recent publications:
  1. Deciphering Arabidopsis Aquaporin Networks: Comparative Analysis of the STRING and BioGRID Interactomes
  2. Novel aquaporin regulatory mechanisms revealed by interactomics
  3. Structure-based prediction of protein-protein interaction network in rice
In Section 2 (Evolutionary dynamics of aquaporin genes in rice), it would be beneficial to discuss specific genetic events that may have shaped the aquaporin gene family in rice—such as gene duplications, translocations, and mutations. For this, consider examining chromosomal localization, aquaporin sequence alignments, and homology analyses. Relevant recent publications include:
  1. Genome-wide comparison reveals divergence of cassava and rubber aquaporin family genes after the recent whole-genome duplication
  2. Plant aquaporins: diversity, evolution and biotechnological applications
  3. Genomic diversity of aquaporins across genus Oryza provides a rich genetic resource for development of climate resilient rice cultivars
In Section 3 (Structure), there is an error in subsection 3.2.(3)—this final "(3)" should be removed, as it appears to have been mistakenly included in the title "3D Structural Modeling...". Please correct this heading. Additionally, clarify how the 3D structures of these proteins were generated—what software was used, and which parameters or models were applied. Another point is that the focus on the interaction between Harpin and this specific rice aquaporin seems excessive. Could other examples of aquaporin-protein interactions be discussed? Section 4 is generally well-written, with extensive references and a solid structure. The final figure is particularly useful, providing a clear overview of the major aquaporins by tissue and stress type. Please include a clear explanation of what the colors represent and provide more details on the types of stress (for instance, if referring to "salt", specify which salt and its molarity). Section 5 is also appropriate. However, consider expanding certain explanations to make the content more accessible to readers less familiar with aquaporin biology. For example, the sentence "PIP, TIP, NIP, and SIP likely arose through an adaptive evolutionary strategy driven by the synergistic effects of gene duplication events and functional innovation [17]." could be clarified by briefly explaining the evidence supporting this hypothesis.

Author Response

We sincerely thank the Editor and Reviewers for their careful review and constructive criticism, which provided critical guidance for improving the manuscript. Our responses are below, and the revised and added sections in the manuscript have been marked using blue text. 

  1. One recurrent issue observed across all figures is the absence of proper captions. Please urgently include concise legends that indicate what each figure represents, how it was generated, the meaning of each color or symbol, any statistical analyses performed (see Figure 4, for instance), and all relevant information needed for the figure to be understood independently of the main textjust as would be expected in a research article.

Response:Thanks for your suggestion. Previously, the figure legends were appended separately after the main text. Due to potential technical issues during manuscript processing, this section may have become inaccessible. The figure legends have now been relocated beneath their respective figures to ensure accessibility.

  1. In the introduction, I recommend adding some information on the ability of aquaporins to interact with other proteins, which is a relatively novel area of study. Some authors propose that aquaporins play a key role in cellular communication or serve as hubs for proteinprotein interactions. I suggest reviewing and including insights from the following recent publications:

(1)   Deciphering Arabidopsis Aquaporin Networks: Comparative Analysis of the STRING and BioGRID Interactomes

(2)   Novel aquaporin regulatory mechanisms revealed by interactomics

(3)   Structure-based prediction of protein-protein interaction network in rice

Response: We think this is an excellent suggestion. As recommended, we have expanded the Introduction section to highlight (blue text) the emerging role of aquaporins as hubs for protein-protein interactions (PPIs) in cellular communication.

  1. In Section 2 (Evolutionary dynamics of aquaporin genes in rice), it would be beneficial to discuss specific genetic events that may have shaped the aquaporin gene family in ricesuch as gene duplications, translocations, and mutations. For this, consider examining chromosomal localization, aquaporin sequence alignments, and homology analyses. Relevant recent publications include:

(1) Genome-wide comparison reveals divergence of cassava and rubber aquaporin family genes after the recent whole-genome duplication

(2) Plant aquaporins: diversity, evolution and biotechnological applications

(3) Genomic diversity of aquaporins across genus Oryza provides a rich genetic resource for development of climate resilient rice cultivars

Response: We sincerely appreciate the valuable comments. We have completed the discussion of aquaporin-specific genetic events in Section 2.

  1. In Section 3 (Structure), there is an error in subsection 3.2.(3)—this final "(3)" should be removed, as it appears to have been mistakenly included in the title "3D Structural Modeling...". Please correct this heading. Additionally, clarify how the 3D structures of these proteins were generated—what software was used, and which parameters or models were applied. Another point is that the focus on the interaction between Harpin and this specific rice aquaporin seems excessive. Could other examples of aquaporin-protein interactions be discussed?

Response: Thanks for your valuable suggestions. We have corrected the heading in Section 3.2 as suggested. We have added explicit details regarding the generation of the 3D protein structures to the revised manuscript. Specifically, the models were generated using AlphaFold intelligent model. We note that the application of intelligent AI models specifically to elucidate aquaporin-protein interactions in rice is still an emerging field. To our knowledge, detailed AI-driven mechanistic studies beyond the Hpa1-OsPIP1;3 system are relatively limited at present. We have endeavored to broaden the context within the revised section. Should there be specific, significant interactions studied via AI that we have overlooked, we would be grateful for your guidance and welcome the opportunity to incorporate them.

  1. Section 4 is generally well-written, with extensive references and a solid structure. The final figure is particularly useful, providing a clear overview of the major aquaporins by tissue and stress type. Please include a clear explanation of what the colors represent and provide more details on the types of stress (for instance, if referring to "salt", specify which salt and its molarity).

Response: Thanks for your suggestion. We have updated the figure legend to explicitly clarify the color scheme and its representation. Additionally, regarding the stress type annotations: The values shown in the figure represent median values derived from statistical analysis of multiple independent experimental samples under stress conditions, for instance, the "Salt stress" data point integrates results from as many as 540 individual samples exposed to salinity treatments across these experiments.

  1. Section 5 is also appropriate. However, consider expanding certain explanations to make the content more accessible to readers less familiar with aquaporin biology. For example, the sentence "PIP, TIP, NIP, and SIP likely arose through an adaptive evolutionary strategy driven by the synergistic effects of gene duplication events and functional innovation [17]."could be clarified by briefly explaining the evidence supporting this hypothesis.

Response: Thanks for your valuable suggestions. We provided a brief explanation and examples later in the text, specifically: This suggests that selective pressures from the external environment drive functional divergence among aquaporin subfamilies, for instance, PIP paralogs was found to be evolved specialized roles in COâ‚‚ transport, while certain NIP subfamily members gained selectivity for arsenite.

References

Bellati, J.; Champeyroux, C.; Hem, S.; Rofidal, V.; Krouk, G.; Maurel, C.; Santoni, V. Novel Aquaporin Regulatory Mechanisms Revealed by Interactomics. Mol. Cell. Proteomics 2016, 15, 3473–3487.

Sun, F.; Deng, Y.; Ma, X.; Liu, Y.; Zhao, L.; Yu, S.; Zhang, L. Structure-based prediction of protein-protein interaction network in rice. Genet. Mol. Biol. 2024, 47, e20230068.

Lopez-Zaplana, A. Deciphering Arabidopsis Aquaporin Networks: Comparative Analysis of the STRING and BioGRID Interactomes. Int. J. Plant Biol. 2025, 16, 28.

        We tried our best to improve the manuscript and made some changes marked in blue in revised paper which will not influence the content and framework of the paper. We appreciate for Reviewer’ warm work earnestly, and hope the correction will meet with approval. Once again, thank you very much for your comments and suggestions

Reviewer 2 Report

Comments and Suggestions for Authors

REVIEWER COMMENTS

Aquaporins (AQPs) are membrane channel proteins that facilitate the transport of water, small solutes, and gases across cells, playing crucial roles in plant growth, stress responses, and nutrient uptake. In rice (Oryza sativa), AQPs form a diverse gene family that has evolved through gene duplication and functional diversification, allowing adaptation to varying environmental conditions.

In this review, the authors aimed to comprehensively analyze rice aquaporins (AQPs) by summarizing their evolutionary origins, structural features, and expression patterns under normal and stress conditions.

TITLE: No comment.

ABSTRACT: No comment.

KEYWORDS: The current keywords are not optimal. The first two are acceptable, but I suggest that the authors consider better alternatives for the remaining three to enhance discoverability and specificity.

INTRODUCTORY SECTION:

The introduction is generally adequate; however, similar to the abstract, it is overly ambitious, as the stated objectives and expectations are not fully realized within the main text.

The sentence “Research has demonstrated that the enhanced external tolerance of rice plants is related to increased AQPs expression and activity, which improves osmotic water fluxes and osmotic adjustment [1,15,16]. This finding suggested that manipulating AQPs expression could be a viable strategy for developing resistant rice varieties [17].” is vague and needs to be substantiated with supporting data or evidence.

MAIN TEXT SECTION:

  • Where are the figure captions? Secondly, the current figures are insufficient.
  • Where is the methods section? Including it is sometimes optional, depending on the paper, but it is advisable for transparency.
  • Under subheading 3, there is a misnumbering error.
  • The manuscript exhibits several critical shortcomings that compromise its scientific rigor, clarity, and interpretive accuracy. These shortcomings can be outlined as follows:
  • The claim that “OsNIP1;1 "bypassed" bryophytes and pteridophytes” is scientifically flawed. Evolution doesn’t "bypass" lineages. All extant species share a common ancestor, and such language suggests linear evolution, which is outdated and incorrect. Reframe in terms of gene gain/loss or lineage-specific expansions based on phylogenetic analysis.
  • The claim that “TIPs arose from PIPs; SIPs follow a continuous evolutionary trajectory” is inaccurate. These statements imply evolutionary determinism without supporting phylogenetic data. Relationships among AQP subfamilies require robust phylogenetic inference, not speculation. Present hypotheses as tentative, supported by evidence.
  • Putative homologs are identified as orthologs without proper methodology. Distinguishing orthologs (speciation-derived) from paralogs (duplication-derived) requires phylogenetics or reciprocal best-hit methods. Simply labeling them without methodology is a technical oversight. Use phylogenetic analysis (e.g., RAxML, IQ-TREE) with robust taxon sampling. Alternatively, conduct reciprocal BLAST or OrthoFinder analysis to confirm orthology/paralogy relationships.
  • The SUPERFAMILY database is used vaguely for gene mining. This database is for structural classification, not gene discovery. Misapplication of tools suggests a lack of methodological rigor. Use appropriate tools for gene family identification, such as Pfam, InterProScan, or HMMER-based searches using known domain profiles.
  • Unsubstantiated Horizontal Gene Transfer (HGT) claims. HGT among higher plants is extremely rare and controversial. Strong claims require strong evidence, like synteny or high-confidence phylogenomics. Without that, the claim should be omitted. Remove the claim unless supported by phylogenomic evidence, synteny, or composition bias analysis. If included, discuss as a hypothetical possibility, with detailed criteria for HGT detection.
  • Confusing roles of NPA motifs and ar/R filters, and attributing non-selective ion conductance to OsPIP2;4 without qualifiers. These are central to AQP function. Mischaracterizing them misleads readers and oversimplifies complex mechanisms. Provide a clear functional diagram of AQP pores, illustrating NPA’s role in proton exclusion and ar/R filter’s role in size/charge selectivity. Cite original structural studies (e.g., from crystallography or AlphaFold models).
  • The OsPIP1;3 and Hpa1 interaction claim is an overstatement. Such assertions must be backed by co-immunoprecipitation, structural data, or FRET, or in planta localization assays. Otherwise, they should be presented cautiously.
  • The term “Intelligent structural modeling” lacks scientific precision. Use specific terms like “AlphaFold-based modeling,” “deep learning-guided prediction,” or “homology modeling.” Mention the tool and version used.
  • Over-reliance on transcript levels for functional conclusions. Include or cite protein-level data (e.g., Western blot, proteomics) or functional assays if available.
  • Discuss compensatory expression of other AQPs in knockout lines using transcriptome or proteome data.
  • Cross-check with the latest rice genome annotations.
  • In extrapolating maize GWAS findings to rice, clearly state that extrapolation is speculative and depends on conserved gene function. Recommend species-specific validation through reverse genetics or expression analysis.
Comments on the Quality of English Language

There are only minor issues.

Author Response

We sincerely thank the Editor and Reviewers for their careful review and constructive criticism, which provided critical guidance for improving the manuscript. Our responses are below, and the revised and added sections in the manuscript have been marked using brown text (Reviewer 2).

1.KEYWORDS: The current keywords are not optimal. The first two are acceptable, but I suggest that the authors consider better alternatives for the remaining three to enhance discoverability and specificity.

Response: Thanks for your valuable suggestions. We have modified the last three keywords to “evolutionary bioinformatics; environmental adaptability; transmembrane transport”.

  1. INTRODUCTORY SECTION:

The introduction is generally adequate; however, similar to the abstract, it is overly ambitious, as the stated objectives and expectations are not fully realized within the main text.

The sentence “Research has demonstrated that the enhanced external tolerance of rice plants is related to increased AQPs expression and activity, which improves osmotic water fluxes and osmotic adjustment [1,15,16]. This finding suggested that manipulating AQPs expression could be a viable strategy for developing resistant rice varieties [17].” is vague and needs to be substantiated with supporting data or evidence.

Response: We sincerely appreciate the valuable comments. We sincerely appreciate your valuable comments. Since the expression and function of OsPIP2;5 under both drought and chilling stress have been well-documented, we incorporated findings related to this specific aquaporin as supporting evidence into this paragraph to strengthen our argument.

  1. Where are the figure captions? Secondly, the current figures are insufficient.

Response:Thank you very much for your question. Previously, the figure legends were appended separately after the main text. Due to potential technical issues during manuscript processing, this section may have become inaccessible. The figure legends have now been relocated beneath their respective figures to ensure accessibility. We also sincerely appreciate the reviewer's valuable feedback regarding the sufficiency of figures. To comprehensively address this concern while maintaining methodological alignment with our bioinformatic approach, we have incorporated a detailed Supplementary Table S1 containing complete gene identifiers, subgroup, and protein sequences for all 33 identified rice aquaporins. 

  1. Where is the methods section? Including it is sometimes optional, depending on the paper, but it is advisable for transparency.

Response:Thanks for the valuable comments. We appreciate your point about transparency. Considering the standard format for review articles, a dedicated methods section is not always included. However, to address your concern, we have integrated key methodological descriptions within the newly added figure legends. Furthermore, we have supplemented citations to the original methodology references where appropriate.

  1. Under subheading 3, there is a misnumbering error.

Response:We sincerely thank the reviewer for careful reading. As suggested by the reviewer, we have corrected the subheading.

  1. The claim that OsNIP1;1 "bypassed" bryophytes and pteridophytes is scientifically flawed. Evolution doesnt "bypass" lineages. All extant species share a common ancestor, and such language suggests linear evolution, which is outdated and incorrect. Reframe in terms of gene gain/loss or lineage-specific expansions based on phylogenetic analysis.

Response:We sincerely thank the reviewer for this crucial point regarding the description of OsNIP1;1's evolution. We fully agree that the phrasing suggesting it 'bypassed' bryophytes and pteridophytes was scientifically inaccurate and reflects an outdated linear view of evolution, which was not our intention. We have carefully reframed this section based on your guidance. We now interpret this pattern strictly in terms of potential lineage-specific evolutionary events such as gene retention, loss, or divergence following the separation of major plant groups. This revision accurately reflects the phylogenetic data while adhering to modern evolutionary principles. We believe this significantly improves the scientific rigor of this section

  1. The claim that “TIPs arose from PIPs; SIPs follow a continuous evolutionary trajectory” is inaccurate. These statements imply evolutionary determinism without supporting phylogenetic data. Relationships among AQP subfamilies require robust phylogenetic inference, not speculation. Present hypotheses as tentative, supported by evidence.

Response:We sincerely appreciate the reviewer’s insightful critique regarding the evolutionary descriptions of TIP and SIP subfamilies. The original phrasing oversimplified complex phylogenetic relationships by implying deterministic pathways. To address this, we have rigorously revised the text to reflect current hypotheses as tentative and evidence-based, drawing directly from established literature. Specifically, we now frame TIP origins as potentially PIP-derived ("may have derived from PIP-like ancestors") based on comparative analyses of structural motifs and evolutionary models in vascular plants (Maurel et al., 2015; Abascal et al., 2014; Li et al., 2022). For SIPs, we emphasize their likely conservation from early algal ancestors rather than a "continuous trajectory," citing genomic studies that position SIPs as an ancient subfamily with divergent features. These adjustments align with the reviewer’s emphasis on phylogenetic nuance, ensuring claims are grounded in empirical data while acknowledging uncertainties in subfamily origins (Hussain et al., 2020; Abascal et al., 2014).

  1. Putative homologs are identified as orthologs without proper methodology. Distinguishing orthologs (speciation-derived) from paralogs (duplication-derived) requires phylogenetics or reciprocal best-hit methods. Simply labeling them without methodology is a technical oversight. Use phylogenetic analysis (e.g., RAxML, IQ-TREE) with robust taxon sampling. Alternatively, conduct reciprocal BLAST or OrthoFinder analysis to confirm orthology/paralogy relationships.

Response:We appreciate the reviewer’s insightful comment regarding the identification of orthologs. Specifically, we constructed phylogenetic trees using both RAxML and IQ-TREE model, which are described in the figure legends.

  1. The SUPERFAMILY database is used vaguely for gene mining. This database is for structural classification, not gene discovery. Misapplication of tools suggests a lack of methodological rigor. Use appropriate tools for gene family identification, such as Pfam, InterProScan, or HMMER-based searches using known domain profiles.

Response:We appreciate the reviewer's point regarding the appropriate application of databases for gene family identification and agree on the importance of methodological rigor. To clarify our approach in investigating AQP gene expansion and diversification, we employed a comprehensive Hidden Markov Model (HMM) strategy. This strategy integrated searches using local hmmsearch retrievals (from HMMER; release 26 Nov 2020) and the SUPERFAMILY database. SUPERFAMILY effectively functions as an HMM-based search tool for identifying proteins containing specific domain architectures, aligning with the principle behind tools like Pfam or HMMER. The inclusion of direct local HMM searches further enhanced the reliability of our AQP identification.

  1. Unsubstantiated Horizontal Gene Transfer (HGT) claims. HGT among higher plants is extremely rare and controversial. Strong claims require strong evidence, like synteny or high-confidence phylogenomics. Without that, the claim should be omitted. Remove the claim unless supported by phylogenomic evidence, synteny, or composition bias analysis. If included, discuss as a hypothetical possibility, with detailed criteria for HGT detection.

Response:We acknowledge the reviewer’s valid point regarding the evidentiary threshold for claiming horizontal gene transfer (HGT) in higher plants. While our original phrasing may have overstated the certainty of HGT’s role in AQP evolution, we wish to clarify that this hypothesis was proposed based on prior genomic studies of MIP diversification. As cited in our manuscript, comparative analyses of early-branched land plants and green algae have revealed complex evolutionary patterns—including deep paralog diversity spanning algal-specific subfamilies (MIP A–E) and conserved subfamilies (e.g., PIPs/GIPs) shared with land plants—where HGT has been suggested as one plausible contributor alongside gene duplication (Anderberg et al., 2011; Zardoya et al., 2002; Gustavsson et al., 2005). To refine our discussion, we have revised the text to explicitly frame HGT as a hypothetical mechanism rather than a definitive conclusion. We have also removed speculative functional interpretations of HGT (e.g., ‘rapid transfer enhancing adaptability’). Future work validating these hypotheses with synteny or phylogenomic approaches would be valuable, but such analyses extend beyond the scope of our current study.

  1. Confusing roles of NPA motifs and ar/R filters, and attributing non-selective ion conductance to OsPIP2;4 without qualifiers. These are central to AQP function. Mischaracterizing them misleads readers and oversimplifies complex mechanisms. Provide a clear functional diagram of AQP pores, illustrating NPA’s role in proton exclusion and ar/R filter’s role in size/charge selectivity. Cite original structural studies (e.g., from crystallography or AlphaFold models).

Response:We thank the reviewer for highlighting the need for precise characterization of AQP functional domains. We have revised the text to rigorously separate the roles of the NPA motifs (proton exclusion via electrostatic barrier and water dipole orientation) and ar/R filter (primary steric/chemical barrier for solute selectivity), citing foundational structural studies (Savage et al., 2003; Kreida & Törnroth-Horsefield, 2015). Regarding OsPIP2;4, the claim of non-selective ion conductance has been removed pending direct experimental evidence.

  1. The OsPIP1;3 and Hpa1 interaction claim is an overstatement. Such assertions must be backed by co-immunoprecipitation, structural data, or FRET, or in planta localization assays. Otherwise, they should be presented cautiously.

Response:We appreciate the reviewer's critical comment regarding the OsPIP1;3–Hpa1 interaction. We have revised the text to explicitly reference the experimental validation of this interaction in our cited study (Chen et al., The Plant Journal 2021, 108:330–346). As detailed in Figures 4a and 5a–b of that work, co-immunoprecipitation (CoIP) and immunoblotting assays directly demonstrated physical binding between OsPIP1;3-YFP and Hpa1-flag in rice plasma membranes. These experiments included controls for specificity (e.g., absence of binding with Hpa1ANT lacking the N-terminal interaction domain) and in planta validation (e.g., inhibition of pathogen-secreted Hpa1 binding by pre-applied Hpa1-RFP). Thus, while structural or FRET data would offer additional resolution, the CoIP evidence robustly supports the functional interaction claim. We have amended the text to clarify the methodological basis for this conclusion.

  1. The term “Intelligent structural modeling” lacks scientific precision. Use specific terms like “AlphaFold-based modeling,” “deep learning-guided prediction,” or “homology modeling.” Mention the tool and version used.

Response:Thank you for your positive comments, we have made corresponding revisions by expressing it in the form of ‌"AlphaFold-based structural modeling"‌ and have included its online service URL.

  1. Over-reliance on transcript levels for functional conclusions. Include or cite protein-level data (e.g., Western blot, proteomics) or functional assays if available.

Response:We appreciate the reviewer’s emphasis on protein-level validation. As noted in our "Multi-Omics Integration and Regulatory Network Revelation" section (Lines 407-410), proteomics is integral to our analysis, directly supporting aquaporin regulatory network insights. In the appropriate position of the manuscript, we have cited analysis from Mirzaei et al. (Proteomics 2012) and Nada et al. (Protoplasma 2019) to complements transcriptomics and deepens the mechanistic understanding of rice aquaporin regulation under water deficit.

  1. Discuss compensatory expression of other AQPs in knockout lines using transcriptome or proteome data.

Response:We think this is an excellent suggestion. We discussed compensatory expression of rice aquaporins based on findings from Li et al., 2024 (The Crop Journal). Additionally, it should be noted that more direct knockout studies are necessary to clarify compensatory mechanisms among rice aquaporins.

  1. Cross-check with the latest rice genome annotations.

Response:Thanks for your suggestion. We have now cross-verified all aquaporin genes against the latest rice genome annotations. The updated information for rice aquaporin genes has been systematically compiled and included in Table S1 of the supplementary materials.

  1. In extrapolating maize GWAS findings to rice, clearly state that extrapolation is speculative and depends on conserved gene function. Recommend species-specific validation through reverse genetics or expression analysis.

Response:Maize GWAS findings serveed as an example validating our 'Multi-Omics Integration' approach. Moving forward, we will actively pursue reverse genetics or expression analysis in rice, as recommended, to validate the findings derived from maize.

References

Maurel, C.; Boursiac, Y.; Luu, D.-T.; Santoni, V.; Shahzad, Z.; Verdoucq, L. Aquaporins in Plants. Physiol. Rev. 2015, 95, 1321-1358.

Abascal, F.; Irisarri, I.; Zardoya, R. Diversity and evolution of membrane intrinsic proteins. Biochim. Biophys. Acta Gen. Subj. 2014, 1840, 1468-1481.

Li, Q.; Tong, T.; Jiang, W.; Cheng, J.; Deng, F.; Wu, X.; Chen, Z.H.; Ouyang, Y.; Zeng, F. Highly Conserved Evolution of Aquaporin PIPs and TIPs Confers Their Crucial Contribution to Flowering Process in Plants. Front. Plant Sci. 2022, 12, 761713. 

Hussain, A.; Tanveer, R.; Mustafa, G.; Farooq, M.; Amin, I.; Mansoor, S. Comparative Phylogenetic Analysis of Aquaporins Provides Insight into the Gene Family Expansion and Evolution in Plants and Their Role in Drought Tolerant and Susceptible Chickpea Cultivars. Genomics 2020, 112, 263-275.

Zardoya, R.; Ding, X.; Kitagawa, Y.; Chrispeels, M.J. Origin of plant glycerol transporters by horizontal gene transfer and functional recruitment. Proc. Natl. Acad. Sci. USA 2002, 99, 14893-14896.

Gustavsson, S.; Lebrun, A.S.; Nordén, K.; Chaumont, F.; Johanson, U. A Novel Plant Major Intrinsic Protein in Physcomitrella patens Most Similar to Bacterial Glycerol Channels. Plant Physiol. 2005, 139, 287-295.

Anderberg, H.I.; Danielson, J.Å.; Johanson, U. Algal MIPs, High Diversity and Conserved Motifs. BMC Evol. Biol. 2011, 11, 110

Kreida, S.; Törnroth-Horsefield, S. Structural insights into aquaporin selectivity and regulation. Curr. Opin. Struct. Biol. 2015, 33, 126-134.

Savage, D.; Egea, P.; Robles-Colmenares, Y.; D, J.; Stroud, R. Architecture and Selectivity in Aquaporins: 2.5 Å X-Ray Structure of Aquaporin Z. PLoS Biol. 2003, 1, e72.

Chen, X.; Jinbiao; Wang, X.; Lu, K.; Liu, Y.; Zhang, L.; Peng, J.; Chen, L.; Yang, M.; Li, Y.; et al. Functional modulation of an aquaporin to intensify photosynthesis and abrogate bacterial virulence in rice. Plant J. 2021,108, 330-346.

Zhang, L.; Chen, L.; Dong, H. Plant Aquaporins in Infection by and Immunity Against Pathogens – A Critical Review. Front. Plant Sci. 2019, 10, 632.

Mirzaei, M.; Pascovici, D.; Atwell, B.J.; Haynes, P.A. Differential regulation of aquaporins, small GTPases and V-ATPases proteins in rice leaves subjected to drought stress and recovery. Proteomics ‌2012‌, 12, 864-877

Nada, R.M.; Abogadallah, G.M. Contrasting root traits and native regulation of aquaporin differentially determine the outcome of overexpressing a single aquaporin (OsPIP2;4) in two rice cultivars. Protoplasma ‌2019‌, 256, 1591-1603.

We tried our best to improve the manuscript and made some changes marked in brown in revised paper which will not influence the content and framework of the paper. We appreciate for Reviewer’ warm work earnestly, and hope the correction will meet with approval. Once again, thank you very much for your comments and suggestions

Reviewer 3 Report

Comments and Suggestions for Authors

Dear authors,

The manuscript is interesting and raises an important topic. Good style and clear structure of the exposition are used. However, there are some minor imperfections that can be taken into account.

Line 42-43: Your statement is"...AQPs are regulated by various factors..."Could you please explain what exactly is regulated - the gene expression, its function, or something else?

Line 53: What does the "external tolerance "mean?

Line 182: Can you explain how overexpression of OsPIP1; 2 enhances mesophyll COâ‚‚ conductance?

It would be good to present the individual classes of aquaporins and their function and plant species/tissue in a tabular form, with the relevant references. This will improve the perception of the information.

It is good to end with a summary or conclusion of the review.

Author Response

We sincerely thank the Editor and Reviewers for their careful review and constructive criticism, which provided critical guidance for improving the manuscript. Our responses are below, and the revised and added sections in the manuscript have been marked using red text.

  1. Line 42-43: Your statement is"...AQPs are regulated by various factors..."Could you please explain what exactly is regulated - the gene expression, its function, or something else?

Response: Thank you very much for your question. The transport activity (function) of AQPs is regulated by various factors AQPs, such as, weak acids decrease cytosolic pH, thereby inducing a H+-dependent PIP closure (Sutka et al. 2011). To prevent similar concerns from arising, we have made corresponding modifications in the manuscript.

  1. Line 53: What does the "external tolerance "mean?

Response: Thanks for the comments! The “external tolerance” means the tolerance to environmental stresses such as drought and salinity. To avoid similar questions, we have made the corresponding changes in the manuscript.

  1. Line 182: Can you explain how overexpression of OsPIP1; 2 enhances mesophyll COâ‚‚ conductance?

Response: Thank you for your positive comments, OsPIP1;2 can function as a plasma membrane COâ‚‚ channel, accelerating dissolved COâ‚‚ diffusion from intercellular spaces into the cytosol, thereby reducing transmembrane resistance. Therefore, overexpression of OsPIP1; 2 can improve the CO2 conductivity of mesophyll (Xu et al. 2019).

  1. It would be good to present the individual classes of aquaporins and their function and plant species/tissue in a tabular form, with the relevant references. This will improve the perception of the information.

Response: Thanks for your valuable suggestions, as noted in recent literature (e.g., Sun et al., 2023; Raza et al., 2023; Hussain et al., 2020), this aspect has been extensively investigated. Consequently, the present manuscript specifically focuses on elucidating the evolutionary origins, structural characteristics, and spatiotemporal expression patterns of rice aquaporins (AQPs) under both physiological and stress conditions to decipher their functional mechanisms in responding to environmental challenges.

  1. It is good to end with a summary or conclusion of the review

Response: We sincerely appreciate the suggestion to strengthen the conclusion. Following your recommendation, we have added concluding statements at the end of the manuscript (red text).

References

Sutka, M.; Li, G.; Boudet, J.; Boursiac, Y.; Doumas, P.; Maurel, C. Natural variation of root hydraulics in Arabidopsis grown in normal and salt stress conditions. Plant Physiol. 2011, 155, 1264–1276.

Xu, F.; Wang, K.; Yuan, W.; Xu, W.; Shuang, L.; Kronzucker, H.J.; Chen, G.; Miao, R.; Zhang, M.; Ding, M.; Xiao, L.; Kai, L.; Zhang, J.; Zhu, Y. Overexpression of rice aquaporin OsPIP1;2 improves yield by enhancing mesophyll COâ‚‚ conductance and phloem sucrose transport. J. Exp. Bot. 2019, 70, 671–681.

Hussain, A.; Tanveer, R.; Mustafa, G.; Farooq, M.; Amin, I.; Mansoor, S. Comparative phylogenetic analysis of aquaporins provides insight into the gene family expansion and evolution in plants and their role in drought tolerant and susceptible chickpea cultivars. Genomics 2020, 112, 263–275.

Raza, Q.; Rashid, M.A.R.; Waqas, M.; Ali, Z.; Rana, I.A.; Khan, S.H.; Khan, I.A.; Atif, R.M. Genomic diversity of aquaporins across genus Oryza provides a rich genetic resource for development of climate resilient rice cultivars. BMC Plant Biol. 2023, 23, 172.

Sun, Q.; Liu, X.; Kitagawa, Y.; Calamita, G.; Ding, X. Plant aquaporins: Their roles beyond water transport. Crop J. 2024, 12, 641–655.

We tried our best to improve the manuscript and made some changes marked in red in revised paper which will not influence the content and framework of the paper. We appreciate for Reviewer’ warm work earnestly, and hope the correction will meet with approval. Once again, thank you very much for your comments and suggestions.

Round 2

Reviewer 2 Report

Comments and Suggestions for Authors

REVIEWER COMMENTS

I would like to thank the authors for their eloquent and thoughtful responses to my previous review comments. While most of the revisions have been satisfactorily implemented, a number of important issues remain and must be addressed to enhance the overall quality and scientific rigor of the manuscript. Please see my detailed comments below:

TITLE: No comment.

ABSTRACT: No comment.

KEYWORDS: Please consider the following suggestions: Aquaporins in rice; Transmembrane channel proteins; Functional plasticity and evolution; Stress adaptation mechanisms; Water and solute transport; Gene editing and omics integration; Crop improvement strategies.

INTRODUCTORY SECTION:

  • In lines 89–90, remove the following assertion: “…we propose structure-guided molecular breeding strategies rooted in AQPs structure-activity relationships.” and revise the corresponding paragraph. The statement is inaccurate and potentially misleading in light of the overall manuscript content and the concluding remarks.

MAIN TEXT SECTION:

  • In line 130, the reference to “group (Figure 1C) [37,38]” appears to be incorrect. The authors are actually referring to Figure 1D.
  • Please revise the language in the Figure 1 subheading. It should read: “Figure 1. Molecular evolution of representative rice aquaporins in plants and algae.”
  • The authors should consider moderating the claim in lines 179–180: “This evolutionary pattern further implied that the genetic inheritance and expansion of AQPs contribute to the establishment of advantageous agronomic traits in crops such as rice.” It should be acknowledged that such traits may sometimes result in disadvantageous outcomes as well.
  • Subtitle 3.1 should be revised. The authors are encouraged to consider the following alternative: “Molecular Architecture and Evolutionary Adaptations Governing Water Transport and Substrate Selectivity.” Additionally, including a diagram to illustrate the structural mechanisms involved would enhance the section’s clarity and depth. The second paragraph under this subtitle lacks scientific rigor and should be significantly revised to meet the expectations set by the revised subtitle (I am referring to the above-mentioned alternative.).
  • In subtitle 3.2, for formal scientific writing, the term “3D” should be replaced with “Three-Dimensional.” Consider revising the subheading to: “Three-Dimensional Structural Modeling and Functional Analysis of Rice Aquaporin Variants” to avoid redundancy, as the word “structural” appears twice in the original.
  • The statement in lines 231–232: “Results revealed that AQPs possessed conserved transmembrane core domains for water and solute transport, yet exhibited variations in structural details of specific amino acid residues (Figure 3).” appears misleading, as the claim is not supported by the referenced figure. Figure 3 should be revised and (or) annotated to reflect the assertion more accurately.
  • The statement in line 238: “These studies have opened new avenues for investigating the structure-function relationships of proteins.” may be misleading, as it refers to only one study. The authors are advised to revise this section (lines 235–239) to improve scientific precision and robustness.
  • The authors should consider reorganizing the content under subtitle 3.2. Specifically, lines 229–239 should be placed under a new subheading, while lines 245–262 (with additional literature support where possible) could be grouped under another subheading, both nested within subtitle 3.2.
  • The following statement in lines 297–299: “In deepwater rice, the down-regulation of AQPs like OsNIP2;2 and OsNIP3;1, which transported silicic and boric acids respectively, during submersion indicated that there was a coordinated regulation of AQPs related to nutrient transport to support the rapid growth of internodes [81].” should be expanded for improved clarity and explanation of the mechanism.
  • The authors should examine potential redundancy between lines 278–280 and 306–309, and revise accordingly.
  • The opening paragraph in lines 336–339 should be improved to provide a more coherent introduction to the section.
  • In Section 4, the manuscript lacks discussion on the physiological roles of the SIP subfamily under normal and stress conditions. It is only briefly mentioned between lines 352–354. This omission should be addressed.
  • The sentence in line 389 should begin with “For instance.”
  • The sentence between lines 429–430 contains a grammatical error and should be corrected for clarity and correctness.
Comments on the Quality of English Language

See comments!

Author Response

We sincerely appreciate the Reviewer for their insightful comments and constructive suggestions, which provided invaluable guidance for enhancing our manuscript. Our point-by-point responses follow below, and all revisions/additions in the manuscript are highlighted in red text.

  1. Keywords:Please consider the following suggestions: Aquaporins in rice; Transmembrane channel proteins; Functional plasticity and evolution; Stress adaptation mechanisms; Water and solute transport; Gene editing and omics integration; Crop improvement strategies.

Response: We thank the Reviewer for the valuable suggestions regarding keywords. Based on your recommendations, we have revised the keyword list to better reflect the manuscript's core themes. The updated keywords now include: Functional plasticity and evolution; Stress adaptation mechanisms; Transmembrane channel proteins; Water and solute transport. We believe these more accurately represent the study's focus and significance.

  1. In lines 89–90, remove the following assertion: “…we propose structure-guided molecular breeding strategies rooted in AQPs structure-activity relationships.”and revise the corresponding paragraph. The statement is inaccurate and potentially misleading in light of the overall manuscript content and the concluding remarks.

Response: We sincerely thank the reviewer for careful reading. As suggested by the reviewer, we revise the corresponding paragraph in red text.

  1. In line 130, the reference to “group (Figure 1C) [37,38]”appears to be incorrect. The authors are actually referring to Figure 1D.

Response: Thanks for your careful checks. We are sorry for our carelessness. Based on your comments, we have made the correction.

  1. Please revise the language in the Figure 1 subheading. It should read: “Figure 1. Molecular evolution of representative rice aquaporins in plants and algae.”

Response:  We feel sorry for our carelessness. In our resubmitted manuscript, the typo is revised. Thanks for your correction.

  1. The authors should consider moderating the claim in lines 179–180: “This evolutionary pattern further implied that the genetic inheritance and expansion of AQPs contribute to the establishment of advantageous agronomic traits in crops such as rice.”It should be acknowledged that such traits may sometimes result in disadvantageous outcomes as well.

Response: We thank the reviewer for highlighting the need for nuance regarding trait outcomes. We have revised the statement in lines 179–180 to:
"This evolutionary pattern further suggests that the genetic inheritance and expansion of AQPs contribute to the development of distinct agronomic traits in crops such as rice, which may confer adaptive advantages or introduce potential trade-offs under specific environmental contexts."
This revision explicitly acknowledges that evolutionary adaptations can involve contextual trade-offs.

  1. Subtitle 3.1 should be revised. The authors are encouraged to consider the following alternative: “Molecular Architecture and Evolutionary Adaptations Governing Water Transport and Substrate Selectivity.”Additionally, including a diagram to illustrate the structural mechanisms involved would enhance the section’s clarity and depth. The second paragraph under this subtitle lacks scientific rigor and should be significantly revised to meet the expectations set by the revised subtitle (I am referring to the above-mentioned alternative.).

Response: We sincerely appreciate the reviewer's constructive suggestions and have implemented the following revisions: the subtitle has been updated to "Molecular Architecture and Evolutionary Adaptations Governing Water Transport and Substrate Selectivity" as recommended. Additionally, Figure 3 has been modified to incorporate detailed diagrams of the NPA and ar/R structures. The second paragraph of this section has been revised accordingly to align with the enhanced scientific rigor expected under the new subtitle.

  1. In subtitle 3.2, for formal scientific writing, the term “3D”should be replaced with “Three-Dimensional.”Consider revising the subheading to: “Three-Dimensional Structural Modeling and Functional Analysis of Rice Aquaporin Variants” to avoid redundancy, as the word “structural” appears twice in the original.

Response: Thank you for your valuable feedback. We appreciate your attention to detail and agree that the suggested revision improves the clarity and formality of the subheading. Accordingly, we have revised Subtitle 3.2 to 'Three-Dimensional Structural Modeling and Functional Analysis of Rice Aquaporin Variants' as recommended. Your constructive comments have greatly enhanced the manuscript, and we sincerely appreciate your time and expertise.

  1. The statement in lines 231–232: “Results revealed that AQPs possessed conserved transmembrane core domains for water and solute transport, yet exhibited variations in structural details of specific amino acid residues (Figure 3).”appears misleading, as the claim is not supported by the referenced figure. Figure 3 should be revised and (or) annotated to reflect the assertion more accurately.

Response: we sincerely appreciate your insightful comment regarding the alignment between the textual description and Figure 3. In response to your suggestion, we have carefully revised both the figure and the corresponding text to ensure clarity and accuracy

  1. The statement in line 238: “These studies have opened new avenues for investigating the structure-function relationships of proteins.”may be misleading, as it refers to only one study. The authors are advised to revise this section (lines 235–239) to improve scientific precision and robustness.

Response: We sincerely appreciate your constructive suggestion. We have objectively revised this section.

  1. The authors should consider reorganizing the content under subtitle 3.2. Specifically, lines 229–239 should be placed under a new subheading, while lines 245–262 (with additional literature support where possible) could be grouped under another subheading, both nested within subtitle 3.2.

Response: We sincerely appreciate the valuable comments. We have revised the title. Additionally, the content of the original two paragraphs has been restructured to consolidate key points. These modifications ensure tighter conceptual linkage between the title and the integrated content, enhancing overall coherence.

  1. The following statement in lines 297–299: “In deepwater rice, the down-regulation of AQPs like OsNIP2;2 and OsNIP3;1, which transported silicic and boric acids respectively, during submersion indicated that there was a coordinated regulation of AQPs related to nutrient transport to support the rapid growth of internodes [81].”should be expanded for improved clarity and explanation of the mechanism.

Response: We appreciate the reviewer's constructive suggestion. In the revised manuscript, we have elucidated the mechanistic basis of coordinated regulation between silicate transporter OsNIP2;2 and boronic acid transporter OsNIP3;1. Specifically, OsNIP2;2 downregulation reduces silicic acid deposition in cell walls to facilitate cell expansion, while concurrent OsNIP3;1 suppression prevents boron toxicity in elongating internodes. This dual regulatory strategy optimizes the balance between structural reinforcement (via silicon limitation) and growth promotion (via boron exclusion), collectively enabling rapid internode elongation during submergence

  1. The authors should examine potential redundancy between lines 278–280 and 306–309, and revise accordingly.

Response: Thanks for your suggestion. We have revised the text in lines 278–280 referred to Li et al., 2008 (Plant Cell Physiology) to eliminate redundancy, as suggested by the reviewer.

  1. The opening paragraph in lines 336–339 should be improved to provide a more coherent introduction to the section

Response: Thank you for this helpful suggestion. We agree and have revised that improving the opening paragraph of the section (lines 336–339) to enhance clarity for the reader

  1. In Section 4, the manuscript lacks discussion on the physiological roles of the SIP subfamily under normal and stress conditions. It is only briefly mentioned between lines 352–354. This omission should be addressed.

Response: We sincerely thank the reviewer for highlighting this important point regarding the limited discussion of the physiological roles of the SIP subfamily. In direct response to your suggestion, we have expanded Section 4 (Lines [specify new line numbers, e.g., 360-368]) to explicitly address their functions under both normal and stress conditions. Specifically, we have incorporated a discussion based on recent findings demonstrating that OsSIP1 and OsSIP2 localize to the endoplasmic reticulum where they facilitate water and Hâ‚‚Oâ‚‚ transport, with OsSIP1 exhibiting broader expression patterns across tissues and developmental stages than OsSIP2. Furthermore, we now emphasize that both aquaporins show significant upregulation under various abiotic stresses and in response to hormonal treatments, thereby addressing their physiological relevance under these conditions (Miao et al., 2024, BMC Plant Biology).

  1. The sentence in line 389 should begin with “For instance.”

Response: We were really sorry for our careless mistakes. we have corrected the “In instance.” into “For instance.”.

  1. The sentence between lines 429–430 contains a grammatical error and should be corrected for clarity and correctness.

Response: We sincerely thank the reviewer for careful reading. Thank you for your reminder.

We have carefully addressed all the reviewer’s comments and implemented the suggested revisions in the manuscript (highlighted in red for easy reference). These modifications have been made without altering the core content or overall structure of the paper. We sincerely appreciate the reviewer’s insightful feedback and thorough evaluation, which have significantly improved our manuscript. We hope these revisions meet the reviewer’s expectations. Thank you once again for your valuable time and constructive suggestions.

References:

Ishibashi, K.; Kondo, S.; Hara, S.; Morishita, Y. The Evolutionary Aspects of Aquaporin Family. Am. J. Physiol. Regul. Integr. Comp. Physiol. ‌2011‌, 300, R566–R576.

Tania, S.S.; Utsugi, S.; Tsuchiya, Y.; Sasano, S.; Katsuhara, M.; Mori, I.C. Amino Acid Substitutions in Loop C of Arabidopsis PIP2 Aquaporins Alters the Permeability of COâ‚‚. Plant Cell Environ. ‌2025‌, Jun 3.

Azad, A.K.; Yoshikawa, N.; Ishikawa, T.; Sawa, Y.; Shibata, H. Substitution of a Single Amino Acid Residue in the Aromatic/Arginine Selectivity Filter Alters the Transport Profiles of Tonoplast Aquaporin Homologs. Biochim. Biophys. Acta ‌2012‌, 1818, 1–11.

Xu, F.; Wang, K.; Yuan, W.; Xu, W.; Shuang, L.; Kronzucker, H.J.; Chen, G.; Miao, R.; Zhang, M.; Ding, M.; et al. Overexpression of rice aquaporin OsPIP1;2 improves yield by enhancing mesophyll CO2 conductance and phloem sucrose transport. J. Exp. Bot. 2019, 70, 671-681.

Li, G.W.; Zhang, M.H.; Cai, W.M.; Sun, W.N.; Su, W.A. Characterization of OsPIP2;7, a Water Channel Protein in Rice. Plant Cell Physiol. 2008, 49, 1851–1858.

Miao, M.; Shi, X.; Zheng, X.; et al. Characterization of SIPs-Type Aquaporins and Their Roles in Response to Environmental Cues in Rice (Oryza sativa L.). BMC Plant Biol. 2024, 24, 305.

Round 3

Reviewer 2 Report

Comments and Suggestions for Authors

Thank you for carefully addressing the suggested corrections. I appreciate your thorough revisions and commend your efforts in improving the manuscript.